# Marine-Derived *Streptomyces sennicomposti* GMY01 with Anti-Plasmodial and Anticancer Activities: Genome Analysis, In Vitro Bioassay, Metabolite Profiling, and Molecular Docking

**DOI:** 10.3390/microorganisms11081930

**Published:** 2023-07-28

**Authors:** Jaka Widada, Ema Damayanti, Mustofa Mustofa, Achmad Dinoto, Rifki Febriansah, Triana Hertiani

**Affiliations:** 1Department of Agricultural Microbiology, Faculty of Agriculture, Universitas Gadjah Mada, Yogyakarta 55281, Indonesia; 2Research Center for Food Technology and Processing, National Research and Innovation Agency (BRIN), Gunungkidul 55861, Indonesia; emad001@brin.go.id; 3Department of Pharmacology and Therapy, Faculty of Medicine, Public Health and Nursing, Universitas Gajah Mada, Yogyakarta 55281, Indonesia; mustofafk@ugm.ac.id; 4Research Center for Applied Microbiology, National Research and Innovation Agency (BRIN), Cibinong 16911, Indonesia; achm026@brin.go.id; 5Faculty of Medicine and Health Sciences, Universitas Muhammadiyah, Yogyakarta 55183, Indonesia; rifki.febriansah@umy.ac.id; 6Faculty of Pharmacy, Universitas Gadjah Mada, Yogyakarta 55281, Indonesia; hertiani@ugm.ac.id

**Keywords:** anticancer, antimalaria, actinobacteria, genome mining, molecular docking, metabolomic, *Streptomyces*

## Abstract

To discover novel antimalarial and anticancer compounds, we carried out a genome analysis, bioassay, metabolite profiling, and molecular docking of marine sediment actinobacteria strain GMY01. The whole-genome sequence analysis revealed that *Streptomyces* sp. GMY01 (7.9 Mbp) is most similar to *Streptomyces sennicomposti* strain RCPT1-4^T^ with an average nucleotide identity (ANI) and ANI based on BLAST+ (ANIb) values of 98.09 and 97.33% (>95%). An in vitro bioassay of the GMY01 bioactive on *Plasmodium falciparum* FCR3, cervical carcinoma of HeLa cell and lung carcinoma of HTB cells exhibited moderate activity (IC_50_ value of 46.06; 27.31 and 33.75 µg/mL) with low toxicity on Vero cells as a normal cell (IC_50_ value of 823.3 µg/mL). Metabolite profiling by LC-MS/MS analysis revealed that the active fraction of GMY01 contained carbohydrate-based compounds, C17H29NO14 (471.15880 Da) as a major compound (97.50%) and mannotriose (C18H32O16; 504.16903 Da, 1.96%) as a minor compound. Molecular docking analysis showed that mannotriose has a binding affinity on glutathione reductase (GR) and glutathione-S-transferase (GST) of *P. falciparum* and on autophagy proteins (mTORC1 and mTORC2) of cancer cells. *Streptomyces sennicomposti* GMY01 is a potential bacterium producing carbohydrate-based bioactive compounds with anti-plasmodial and anticancer activities and with low toxicity to normal cells.

## 1. Introduction

Marine actinobacteria is a great biotechnologically valuable organism with functional diversity, including diversity in the production of secondary metabolites, especially antibiotics [1]. Various marine actinobacteria, especially the genus *Streptomyces*, have been isolated from marine sediments and sponge species [2]. *Streptomyces*, as the main source of new bioactive compounds, is used to produce two-thirds of all currently available antibiotics. It has a very large genome size, between 6.2 and 12.7 Mb, and 5% of its genome is devoted to the synthesis of secondary metabolites [3,4]. In the past 10 years, the discovery of new compounds from marine bacteria, mainly from the actinobacteria group, has led to the discovery of new antibacterial, antifungal, and antimicrobial compounds [5,6,7,8], anticancer compounds [9,10,11], antioxidants [12,13], anti-complement compounds [14] and antipsychotic compounds [15].

Malaria is a critical human disease caused by *Plasmodium* parasite infection transmitted via a female *Anopheles* mosquito bite [16]. Five species of *Plasmodium* that can infect humans are *Plasmodium falciparum*, *P. vivax*, *P. ovale*, *P. malariae*, and *P. knowlesi* [17]. This infectious disease still has a resistance problem toward first-line drugs for malaria treatment such as sulfadoxine–pyrimethamine, chloroquine, and artemisinin [18,19]. The emergence of drug-resistant *Plasmodium* spp. has driven efforts to find and develop new drugs. The discovery of new antimalarial agents from marine actinobacteria is still limited because 72% of natural products for antimalarial drugs are predominantly derived from plants [20]. New antimalarial drugs could be developed from existing drugs such as anticancer or antibiotic compounds. Previous studies have shown that artemisinin and its derivatives work, as antimalarial drugs have an anticancer effect [19,21]. In another study, an anticancer compound, histone deacetylase inhibitor SB939, inhibited *P. falciparum* and *P. berghei* and showed antimalarial effects [22].

The study of antimalarial and anticancer drug candidates derived from carbohydrate-based compounds is still limited. Previous studies showed that adding functional hydrophilic groups (water-soluble taxoids) in the form of sugars (glucose, galactose, and mannose) can increase solubility and antitumor activity [23]. In recent years, carbohydrate-based compounds have been studied and developed for various drug candidates, such as anticancer, antiviral, antibiotic, and antifungal agents [24]. Anticancer compounds of polysaccharides from *Cyclocarya paliurus* (Batal.) Iljinskaja plant showed a strong inhibition of the growth of human carcinoma HeLa cells [25]. Novel acidic polysaccharides from *Rhynchosia minima* root were found to exhibit anticancer activities against lung cancer A549 and liver cancer HepG2 cells [26]. Fucoidan fractions with galactose residues have an inhibitory effect on colorectal carcinoma cells DLD-1 and HCT-116 [27]. In another study, a carbohydrate-based antibiotic, prumycin, isolated from *Streptomyces* sp. strain F-1028, has anti-plasmodial activities [28]. Moreover, a novel class of carbohydrate-derived thiochromans has antimalarial activity [29,30]. Studies on carbohydrate-based compounds from potential marine actinobacteria, especially *Streptomyces* sp. GMY01, have not been carried out. This study aims to explore potential antimalarial and anticancer compounds derived from *Streptomyces* sp. GMY01 using genome analysis, bioassay, metabolite profiling, and molecular docking.

## 2. Materials and Methods

### 2.1. Biological Materials

Actinobacteria strain GMY01 was isolated from a marine sediment sample collected from Krakal Beach (8°8′44″ S 110°35′59″ E), Gunungkidul, Yogyakarta, Indonesia, as reported in the previous study [31]. It was deposited at the Indonesian Culture Collection (WDCM 769), National Research and Innovation Agency (BRIN), Indonesia, as InaCC A147 and at the NITE Biological Research Center (NBRC, WDCM 825), Japan, with registration number NBRC 110111. This whole-genome shotgun project was deposited at DDBJ/ENA/GenBank under the accession JABBNA000000000 and the assembly accession GCF_012933395.1. *Plasmodium falciparum* FCR3 (chloroquine-resistant strain) was obtained from Eijkman Institute, Jakarta, Indonesia (2018) and was maintained at Pharmacology Laboratory of Faculty of Medicine, Public Health and Nursing Universitas Gadjah Mada (UGM), Yogyakarta, Indonesia. Vero cell line (ATCC-CCL81), HeLa cell line (ATCC CRM-CCLL-2), and HTB cell line were obtained from the same laboratory, UGM, Yogyakarta, Indonesia.

### 2.2. Whole-Genome and Taxonogenomic Analysis

The whole-genome sequencing of *Streptomyces* sp. GMY01 was performed using a combined shotgun sequencing method with the 454 GS FLX Titanium system (Roche) and paired-end (PE) sequencing using the HiSeq 1000 platform (Illumina). The NGS of GMY01 was conducted at Biological Resource Center, NITE (NBRC), Japan, as reported in the previous study [32]. The assembly of the whole-genome sequence was analyzed using Type (Strain) Genome Server (TYGS), a free bioinformatics platform available at https://tygs.dsmz.de (accessed on 18 June 2023), for a whole-genome-based taxonomic analysis, including digital DNA–DNA hybridization (dDDH), G+C content comparison, and phylogeny based on whole-genome sequence [33]. The results were provided by the TYGS on 18 June 2023. The TYGS analysis was divided into the following steps: First, pairwise comparisons of the genome sequence using Genome BLAST Distance Phylogeny (GBDP) and accurate intergenomic distances were inferred using the “trimming” algorithm and distance formula d5 [33]. The dDDH values and confidence intervals were calculated using the recommended settings of genome-to-genome distance calculator (GGDC) 2.1 [33]. The next step was phylogenetic inference. The intergenomic distance obtained was used to arrange the evolutionary tree through FASTME 2.1.6.1, including SPR postprocessing [34]. Branch support was inferred from each of the 100 pseudo-bootstraps. The trees were rooted at the midpoint [35] and visualized with PhyD3 [36], and the final step was grouping species and subspecies using a 70% dDDH radius around each of the 21 type strains, performed as described by Meier-Kolthoff and Goker [33]. The resulting groups are shown in Appendix A. The subspecies were grouped using a 79% dDDH threshold, as previously introduced by Meier-Kolthoff [37]. The average nucleotide identity (ANI) was analyzed using an ANI calculator, which is available online at https://www.ezbiocloud.net/tools/ani (accessed on 18 June 2023) [38]. Among the *Streptomyces* genomes, the correlation of the tetranucleotide signatures (TETRA), the ANI, and the genome-to-genome distance was calculated, and pairwise genomic comparisons were carried out. The statistical calculations of the tetranucleotide frequencies (TETRA) and the ANIb were performed using the JSpecies software tool (version: 4.0.2) available at https://jspecies.ribohost.com/jspeciesws/#analyse (accessed on 18 June 2023) [39]. The recommended species cut-off was 95% for the ANIb and >0.99 for the TETRA signature [40]. The genome was annotated using the rapid annotation using subsystem technology (RAST) server [41,42], available online at https://rast.nmpdr.org/ (accessed on 17 January 2020). The putative secondary metabolites analysis was performed using AntiSMASH version 7.0.0 [43].

### 2.3. Cell Biomass Extraction

*Streptomyces* sp. GMY01 was maintained in an ISP-2 agar medium (Difco, Spark, MD, USA). For the seed medium production, the actinobacteria was cultured for 3 days in a 250 mL Erlenmeyer flask containing 100 mL of Tryptic soy broth (Difco, Spark, USA) under 28 °C and 180 rpm agitation. Then, inoculum was transferred into each flask of the production medium (starch nitrate broth (SNB)). It was then incubated for 11 days in a shaker incubator at 28 °C, at 180 rpm agitation. The SNB medium contained 0.5 g NaCl, 1 g KNO_3_, 0.5 g K_2_HPO_4_, 0.5 g MgSO_4_ 7H_2_O 0.5 g, 0.01 g FeSO_4_ 7H_2_O, and 20 g soluble starch in 1000 mL of distilled water [44]. Cell biomass was separated from the liquid using refrigerated centrifugation at 4137× *g*, 4 °C for 15 min [45]. The cell biomass was extracted using methanol under slow stirring conditions for 30–60 min. The methanolic extract was separated from cells via refrigerated centrifugation. The crude extract was fractionated using *n*-hexane to separate the polar and nonpolar fractions, and then re-evaporated to obtain the dried extract. All chemical reagents and solvents were purchased from Merck KGaA, Darmstadt, Germany.

### 2.4. Antiplasmodial Assay

*Plasmodium falciparum* FCR-3 was cultivated using the previous method with minor modifications [46]. Parasites were maintained in 2% human erythrocytes (red blood cells, RBCs) (O+, male) suspended in RPMI 1640 (Gibco, Thermo Fisher Scientific, Waltham, MA, USA) including 10% human serum (O+, male), and 500 μg of gentamicin (Indofarma, Bekasi, Indonesia) per liter. Flask cultures were incubated in a CO_2_ incubator at 5% CO_2_, 37 °C. Before using the culture for treatment, it was synchronized with 5% sorbitol [47]. For the antiplasmodial assay, the extract was prepared by adding 0.1% dimethyl sulfoxide (DMSO) (Sigma-Aldrich, Burlington, MA, USA) (*w*/*v*) at various concentrations. The *Plasmodium* growth inhibition assay was established in a total volume of 200 μL using 96-well microplates (Iwaki). Each microplate, containing 100 μL of extract solution and 100 μL of *Plasmodium* inoculum at the parasitemia level of 5%, was placed in a 5% CO_2_ incubator (CellXpert C170i, Eppendorf AG, Hamburg, Germany) at 37 °C for 3 days. All treatments were performed in triplicate. The *Plasmodium* growth was observed by making thin blood film preparations with Giemsa coloring, and observations were made under a microscope. The parasitemia was calculated from a minimum of 1000 RBCs. The percent *Plasmodium* inhibition was obtained using the formula:$$\%\;Plasmodium\;inhibition=\frac{A-B}{A}\times 100$$
where A is the parasitemia in control (RPMI medium) and B is the parasitemia in treatment. The IC_50_ values were calculated by GraphPad Prism 9.4.1 by applying nonlinear regression with log (inhibitor) versus normalized response.

### 2.5. Anticancer Assay

A 3-(4,5-dimethylthiazol-2-yl)-2,5-diphenyl tetrazolium bromide (MTT, Sigma-Aldrich, Burlington, MA, USA) assay was employed to evaluate the anticancer and cytotoxic effects of the extract in vitro [48]. HeLa cervical cancer cells and Vero cells, as normal cells, were selected for testing. The Vero cells were grown in Dulbecco’s modified eagle medium–high glucose (DMEM.HG, Gibco, Thermo Scientific, Waltham, MA, USA) supplemented with 10% fetal bovine serum (FBS, Gibco, Thermo Scientific, USA). The HeLa cells were grown in the RPMI 1640 medium supplemented with 10% FBS. The cells were cultured in a sterile flat-bottom 96-well microplate (Iwaki, Tokyo, Japan) at 5 × 10^3^ cells’ density/well and left to adhere overnight at a total volume of 100 μL in a humidified incubator (5% CO_2_, 37 °C). Then, 100 μL of extract solution (in 0.1% DMSO) was added to the cells (0–1000 μg/mL for Vero cells and 0–20 μg/mL for HeLa cells), and the cells were incubated for 24 h before MTT assay. All treatments were performed in triplicate. Liquid medium was aspirated from the plate; 100 μL of 5 mg/mL MTT solution was then added to each well, and the plates were incubated at 37 °C in a CO_2_ incubator for 4 h. Furthermore, 100 μL of DMSO was added to dissolve the formazan crystals. The formazan product was determined using a spectrophotometer at 595 nm on a microplate Elisa reader (Bio-Rad, Hercules, CA, USA). The treated and control absorbance values were reduced by the blank absorbance (medium without cells). The percent cell inhibition was calculated using the formula:$$\%\;cell\;inhibition=\frac{A-B}{A}\times 100$$
where A is the absorbance of control cells and B is the absorbance of treated cells. The IC_50_ values were calculated by GraphPad Prism 9.4.1. by applying nonlinear regression with log (inhibitor) versus normalized response.

### 2.6. Liquid Chromatography–Mass Spectrometry/Mass Spectrometry (LCMS/MS) Analysis

Mass spectrometry analysis was performed on a Xevo G2-XS QTof mass spectrometer (Waters MS Technologies, Milford, CT, USA) [49]. Electrospray ionization was adopted. The scan range was from 100 to 1200 *m*/*z*. The capillary and cone voltages were set at 0.8 kV and 30 kV, respectively, and positive electron spray mode was adopted. The desolvation gas was set to 1000 L/h at a temperature of 500 °C, the cone gas was set to 50 L/h, and the source temperature was set to 120 °C. Ultra-performance liquid chromatography (UPLC) analysis was performed using a Waters Acquity Ultra Performance LC system. Chromatographic separation was carried out on an ACQUITY UPLC HSS T3 column (100 mm × 2.1 mm, 1.7 urn) at a column temperature of 40 °C. The mobile phase consisted of solvent A (0.1% formic acid in water, *v*/*v*) and solvent B (0.1% formic acid in acetonitrile), with gradient polarity (A:B) of 95:0.5 to 0.5:95. The flow rate was set at 0.3 mL/min. The column and auto-sampler were maintained at 40 °C and 20 °C, respectively. The injection volume was 1 μL. The data acquisition and processing were performed using UNIFI. The parameter used was retention time (RT) in the range of 1–16 min.

### 2.7. In Silico Molecular Docking

Molecular docking was applied to predict the binding affinity of several detected compounds on genome mining with proteins that play a role in inhibiting *P. falciparum* and cancer cells. Compound structures were created using the ChemDraw online program (available online https://chemdrawdirect.perkinelmer.cloud/js/sample/index.html (accessed on 28 June 2021)) based on the IUPAC name of the compound in the PubChem database (available online https://pubchem.ncbi.nlm.nih.gov/sources (accessed on 28 June 2021)). This study used target proteins that were obtained from the RCSB database. Lactate dehydrogenase (1CET), glutathione reductase (1ONF), phosphoethanolamine methyltransferase (4FGZ), erythrocyte membrane protein 1 (3CPZ), and glutathione-S-transferase (4ZXG) [50] were used as target proteins of *P. falciparum*. For target proteins in a cancer cells, we used apoptosis proteins BCL-2 (2w3l) and BCL-XL (2yxj) [51] and autophagy-related proteins mTORC1 (6BT0) and mTORC2 (6zwm) [52]. The 2D structure of compounds was retrieved from the ChemSpider webserver and converted to an optimized 3D structure using Marvinsketch tools (https://www.chemaxon.com (accessed on 28 June 2021)). Both proteins and compounds were prepared using auto-dock tools. The proteins were cleaned of water, ions, and other small molecules. Hydrogen atoms were added to polar groups of proteins to minimize errors caused by ionization and tautomeric states of amino acid residues. Molecular docking was performed on Autodock Vina [53], while visualization of binding interactions was performed using the DisCoVery studio visualizer (DS visualizer) tool [54]. The computational simulation was carried out on a Windows 10 Operating system, with an AMD A8 7410 (quad-core; 2.2 GHz) processor and 4 GB of RAM. The molecular docking study was observed from the affinity (kcal/mol) value with a root mean square deviation score less than 2 A, which showed a visualization of the binding interaction between compounds and the active site of proteins.

### 2.8. Ethical Clearance

Ethical approval to conduct the in vitro anti-plasmodial assay was obtained from the Ethics Commission of the Faculty of Medicine, Public Health and Nursing, Universitas Gadjah Mada, Indonesia (Ref. No: KE/FK/0279/EC/2019). Informed consent was obtained from the adult subject who donated the blood samples for culturing the *Plasmodium falciparum* FCR3. All methods were carried out following the Ethics Commission guidelines and regulations (Appendix 02-GL 01-version 1.2 available at http://komisietik.fk.ugm.ac.id/ (accessed on 20 January 2020).

## 3. Results

### 3.1. Whole-Genome and Taxogenomic Analysis

The morphological characteristics of GMY01 on the ISP-2 medium are shown in Figure 1. The GMY01 bacteria featured white spores and a smooth surface. The colony formed a pellet formation in broth media, such as starch nitrate broth (SNB). The GMY01 can produce yellow–brown pigments in agar media and broth medium. This morphology is a character of *Streptomyces* bacteria. The 16S rDNA was analyzed and identified in a previous study as *Streptomyces* species [55].

The full-genome sequencing of strain GMY01 using Illumina methods led to an assembly of 73 contigs for a total genome size of 7.97 Mb. Sequence analysis resulting in species and subspecies clusters generated from genome analysis using TYGS is shown in Appendix A. The taxonomic identification of query strains (GCF_012933395.1_ASM1293339v1_genomic, GMY01) based on the pairwise comparisons method between our genomes versus type-strain genomes is shown in Appendix A. A total of 16 species groups, resulting from the cluster analysis and demand lines provided, were assigned to one of these (strain GMY01) (Figure 2). Based on the phylogenetic analysis of the whole genome, *Streptomyces* sp. GMY01 had the highest dDDH (d4) percentage and GC content difference (%) with *S. sennicomposti* RCPT1-4^T^ (GCA_019890635) (82.3; 0.13) and *S. spiralis* JCM 3302 (GCA_014654675) (32; 1.94) (Appendix A). The ANI and ANIb results showed that *Streptomyces* sp. GMY01 and the closest *Streptomyces* reference strains had OrthuANIu values of 85.61–98.09 and ANIb values of 84.66–97.33 (Table 1). Based on an analysis of genome sequence using the TYGS database and ANI value (>95%), the marine sediment bacterium *Streptomyces* sp. GMY01 belongs to *S. sennicomposti* species.

Organism overview using the SEED viewer (https://rast.nmpdr.org (accessed on 17 January 2020)); [41] showed that the functions of the genes were cataloged into different functional classes (Figure 3). Based on the subsystem in Figure 2, many subsystems appeared to be associated with carbohydrate metabolism, and one of them was galactose uptake and utilization.

Genome mining analysis of GMY01 using antiSMASH version 7.0.0 resulted in 28 biosynthesis gene clusters (BGCs) for putative secondary metabolites (Table 2). The type of BGCS was dominated by polyketide synthase (PKS), non-ribosomal polyketide synthetase (NRPS), and lanthipeptide. Seven putative BGCs have 100% similarity with most similar known clusters.

### 3.2. Anti-Plasmodial and Anticancer Activities

The cell biomass was separated from 8 L of liquid fermentation medium, and 8.77 g of methanolic crude extract was obtained from the biomass. A large amount of yield is extracted from actinobacteria (0.11% of yield). In the anti-plasmodial assay, the polar fraction of methanolic extract of GMY01 was shown to have moderate inhibition against *Plasmodium falciparum* FCR-3, cervical carcinoma of HeLa cells, and lung carcinoma of HTB cells (Figure 4). The IC_50_ value of the polar fraction of methanolic extract was 46.06 μg/mL on *P. falciparum* FCR3, 27.31 μg/mL on Hela cells, and 33.75 μg/mL on HTB cells. This fraction has very low toxicity on mammalian normal cells (Vero cells) with an IC_50_ of 823.3 μg/mL. It was lower than chloroquine (IC_50_ = 234.2 μg/mL) as commercial anti-plasmodial—antimalarial, and doxorubicin (IC_50_ = 86.48 μg/mL) as an anticancer (Figure 4).

### 3.3. Main Constituent Profile and Predicted Galactose Metabolic Pathway

To obtain the active compound profile from the active fraction of GMY01, we employed liquid chromatography–mass spectrometry/mass spectrometry (LC-MS/MS) analysis, and the result is shown in Appendix A. The GMY01 polar bioactive fraction detected three compounds (Table 3). The single major compound (97.50%) was C_17_H_29_NO_14_, with a retention time (RT) of 1.08, and the two minor compounds, mannotriose (C_18_H_32_O_16_) and C_19_H_31_NO_13_, with RTs of 1.07 and 1.64, respectively. The ionization mass spectra of all compounds are shown in Appendix A. Based on the PubChem database (https://pubchem.ncbi.nlm.nih.gov (accessed on 9 November 2020)), the major compound, C_17_H_29_NO_14_, was predicted as N-acetylneuraminyl-(2-6)-galactose (National Center for Biotechnology Information. PubChem Database. CID = 169,679, https://pubchem.ncbi.nlm.nih.gov/compound/N-Acetylneuraminyl-_2-6_-galactose (accessed on 28 April 2020). This compound has a molecular weight of 471.4 g/mol and an exact mass of 471.158805 g/mol, similar to our result. The compound is a combination of two molecular compounds, *N*-acetylneuramic acid, and galactose, which are bound to bond number 2 of *N*-acetylneuramic acid with number 6 of galactose.

The Kyoto Encyclopedia of Genes and Genomes (KEGG) mapped galactose metabolism (Appendix A) of *S. sennicomposti* GMY01; 17 of 65 (26.2%) gene clusters encoded enzymes related to fructose and mannose metabolism, and 13 of 37 (35.1%) gene clusters encoded enzymes related to galactose metabolism. GalK (2.7.1.6), GalT (2.7.7.10), GalE (5.1.3.2) and LacZ_12 (3.2.1.23) served as gene clusters in GMY01 related to galactose metabolism. These gene clusters confirmed that, based on genome analysis, *S. sennicomposti* GMY01 produced N-acetylneuraminyl-(2-6)-galactose (C17H29NO14) as carbohydrate-based compounds. However, although mannotriose is present in the galactose metabolism pathway, the gene clusters involved in mannotriose biosynthesis were not detected in GMY01.

### 3.4. Molecular Docking

In this study, all detected compounds had higher affinity than chloroquine (−4.9 to −5.5 kcal.mol^−1^) as a commercial antimalarial drug (Table 4). The two compounds with the highest affinity were mannotriose (−6.4 to −7.7 kcal.mol^−1^) and N-acetylneuraminyl-(2-6)-galactose (−6.1 to −6.7). Mannotriose had the highest affinity for PMT and EMP1 (−7.7 and −7.6 kcal.mol^−1^), whereas N-acetylneuraminyl-(2-6)-galactose had the highest affinity for GR and LDH of *P. falciparum*. The binding visualization of mannotriose on target proteins of *Plasmodium* is shown in Figure 5. In *Plasmodium* protein, mannotriose bound with protein glutathione reductase (GR) in the form of hydrogen bonds; in protein, lactate dehydrogenase (LDH) was in the form of hydrogen bonds and alkyl, whereas, in phosphoethanolamine methyltransferase (PMT), erythrocyte membrane protein 1 (EMP1) and glutathione-S-transferase (GST), it was predominantly in the form of hydrogen bonds.

On cancer cell proteins, all detected compounds had a lower affinity than doxorubicin (from −7.2 to −10.1 kcal.mol^−1^) as commercial cancer drugs (Table 5). The two detected compounds in MS analysis, mannotriose (from −7.4 to −8.4 kcal.mol^−1^) and N-acetylneuraminyl-(2,6)-galactose (from −6.3 to 8 kcal.mol^−1^), had high affinity against cancer target proteins. The mannotriose had the highest affinity for BCL2 as an apoptosis protein, whereas N-acetylneuraminyl-(2-6)-galactose for mTORC1 is the autophagy protein of cancer cell protein. Overall, mannotriose bound to the four cancer proteins using hydrogen bonds (Figure 6).

## 4. Discussion

The whole-genome sequence (WGS) analysis revealed that *Streptomyces* sp. GMY01 (7.9 Mbp) was most similar to *Streptomyces sennicomposti* RCPT1-4^T^ (Table 1). The *S. sennicomposti* RCPT1-4^T^ derived from the compost of *Senna siamea* (Lam.), collected from an agricultural area in Rayong province, Thailand [56]. The ANI among genes conserved from a pair of genomes was considered a more reliable method than traditional DNA-DNA hybridization [40]. The ANI was calculated by connecting the query genome to 1020 nucleotide fragments, and each approached the subject’s genome [57]. For cases with highly similar genomes, the threshold for assuming two organisms belonging to the same species can be established at value > 94% ANI [40]. In the ANI calculation results (Table 1), the ANI value with one *Streptomyces* reference was above 94%; this indicates that *Streptomyces* sp. GMY01 belongs to the closest species, *S. sennicomposti* RCPT1-4^T^.

The anti-plasmodium activity produced by *S. sennicomposti* GMY01 was suggested to be related to its anticancer activity. Other studies reported that the anticancer compound SB939 had *P. falciparum* anti-plasmodium activity in various phases of the *Plasmodium* life cycle [22]. In addition, the antimalarial compound artemisinin and its analogs showed anticancer activity and have synergistic effects with available anticancer drugs, without increasing toxicity to normal cells (Das 2015). This result was consistent with the hypothesis that anticancer compounds could be anti-plasmodial. However, the mechanism of action, and anti-plasmodial and anticancer properties of GMY01 metabolites need to be further studied.

The GMY01 bioactive fraction was detected to contain three carbohydrate-based compounds (Table 3). Carbohydrate-based active compounds from bacteria have not been widely reported. Based on the bacterial carbohydrate structure database (http://csdb.glycoscience.ru/bacterial/ (accessed on 5 August 2021)), several active carbohydrate compounds synthesized by bacteria were found, as well as other novel disulfides from *Streptomyces* sp., and aminoglycoside antibiotics produced by mutant *Bacillus circulans*. Inositol glycoside, a novel disulfide of 2-(N-acetylcysteinyl) amido-2-deoxy-a-D-glucopyranosyl-*myo*-inositol, was isolated from mycelial extracts of *Streptomyces* sp. AJ 9463 using aqueous methanol [58]. A new aminoglycoside antibiotic, S-11-A, was isolated from the fermentation broth of the 2-deoxystreptamine negative (DOS−) mutant of *Bacillus circulans* S-11 [59]. In another study, a carbohydrate-based antibiotic, prumycin, purified from the cultured broth of *Streptomyces* sp. F-1028, was also found to exhibit anti-plasmodium activities [28]. Moreover, a novel class of carbohydrate-derived thiochromans was found to exhibit antimalarial activity [29,30]. Prumycin, as a water-soluble compound, showed in vitro antimalarial activities against *P. falciparum* strains K1 (drug-resistant) and FCR3 (drug-sensitive), and cytotoxicity against human diploid embryonic cell line MRC-5. The compound also showed in vivo antimalarial activity against rodent malaria-derived strains, *P. berghei* N (drug-sensitive) and *P. yoelii* ssp. NS (chloroquine-resistant) [28]. Other carbohydrate-derived thiochromans have been synthesized, and exhibited antimalarial activity against both the chloroquine-sensitive 3D7 and chloroquine-resistant FCR3 strains, with a nontoxic effect on human fetal lung fibroblast (WI-38) cell line [29].

The N-acetylneuraminyl-(2,6)-galactose that was discovered in bacterial cell biomass marine *S. sennicomposti* GMY01 is an important polar compound because of its anticancer activity, as well as its moderate anti-plasmodium activity. N-acetylneuraminyl-(2,6)-galactose has not been widely reported as having anticancer properties. Galactose is a water-soluble carbohydrate with high hydrophilicity. However, several studies reported that carbohydrates and glycosides showed anticancer activities. A previous study has reported that paclitaxel (Taxol), isolated from the bark of *Taxus brebiforia*, is currently used in the treatment of various kinds of cancers and the synthesized paclitaxel analog galactose-bound taxoid (10-a-GAG-DT) has good water solubility and antitumor activity [23].

Mannotriose’s activity as an anticancer has not been widely reported. However, several studies reported that mannotriose is a compound that comprises one of the important prebiotics, namely mannooligosaccharides (MOS) [60,61], and MOS is predicted to have anticancer activity [62]. In another study, purified MOS yielded 40% of mannobiose and 18% of mannotriose. In cytotoxicity assay, treatment with 500 µg/mL of MOS resulted in 50% cell death of HT29 cells (colon cancer cell line) after 24 h [63]. MOS could also decrease the Caco-2 cell (adenocarcinoma colorectal cell line) viability by 74.19%. At 400 μg/mL, MOS was also shown to be very effective at inhibiting the A549 cell line (lung cancer cell line) [62]. This indicated that mannotriose, which is a constituent of MOS, should be suspected of having anticancer activity even at low–moderate activity levels.

Previous studies have reported several active compounds with a basic structure of carbohydrates that are generally sourced from plants and algae. *Cyclocaryapaliurus* (Batal.) Iljinskaja polysaccharide contains galactose, which is a strong anticancer compound isolated from plants, and exhibits activity in human gastric cancer HeLa cells [25]. In a previous study, three novel acidic polysaccharides comprising arabinose, mannose, glucose, and galactose were isolated from *Rhynchosia minima* root. These compounds, as potential natural anticancer agents, were revealed to have in vitro anticancer properties against lung cancer A549 and liver cancer HepG2 cells [26]. Sulfated and acetylated fucoidan fraction containing fucose, galactose, mannose, glucose, and uronic acid residues were isolated from the brown alga Padina boryana. In C4 galactose residues, a single fucose residue was found. The in vitro study revealed that all fucoidans inhibited cancer cell colony formation at a concentration of 200 μg/mL [27]. The discovery of galactose-bound, naturally water-soluble compounds with a high extract yield (0.11%) from marine *Streptomyces* bacteria biomass was important, as these compounds can be developed as new nontoxic anti-plasmodium and anticancer agents.

The mechanism of carbohydrate-based compounds as anticancer agents has been reported in the literature. Some studies report that their polysaccharide bioactivity is determined by their physicochemical properties (polysaccharide content, type of sugar, and molecular weight). Other studies reported that some polysaccharides with anticancer activity featured free radical scavenging ability, conformation, and specific affinity with cell surface receptors, and antioxidant enzymes [25]. Other polysaccharides’ anticancer agents have different structural characters, especially in terms of molecular weight, monosaccharide composition, and functional group content, which determine their biological activity in fucoidan derivatives [64]. The antifungal studies showed that the prumycin mode of anti-fungal action on *B. cineria* related to the selective inhibition of protein synthesis, and its in vitro antitumor activity on HeLa S3 cells involves both protein and DNA synthesis inhibition [28]. However, there is no clear mechanism for carbohydrate-based anti-plasmodium compounds. One study reported on the antimalarial activity of novel carbohydrate-derived thiochromans against the chloroquine-sensitive (3D7) strain of *P. falciparum*. The high antimalarial activity produced by these compounds was known to be related to important structural features such as the presence of short-chain alkyl substituents, benzyl ether protective groups, and equatorial orientation of C-4 substituents in sugar groups [30].

An in silico molecular docking study showed that N-acetylneuraminyl-(2,6)-galactose and mannotriose had a high affinity for glutathione reductase (GR) (−6.7 and −7.3 kcal.mol^−1^) of *P. falciparum* protein (Table 4). Glutathione is the most abundant low-molecular-weight redox-active thiol in parasites, existing primarily in its reduced form and representing an excellent thiol redox buffer. This allows for the efficient maintenance of the intracellular reducing environment of the parasite cytoplasm and its organelles [65]. Glutathione reductase (GR), which is a major antioxidant enzyme, is targeted in the treatment of many diseases due to the dual role of its product, reduced glutathione (GSH), in infected cells [66]. The affinity of mannotriose was higher than that in another study. Low scores were shown in the docking of resorcinol derivatives with glutathione reductase (GR) protein, which has a docking score between −2.9 and −4.2 kcal/mol [66]. The high affinity of mannotriose to these proteins was predicted this compound to inhibit the growth of *P. falciparum*.

In *P. falciparum* protein, mannotriose and N-acetylneuraminyl-(2,6)-galactose also had a higher affinity to the erythrocyte membrane protein (EMP) (−7.6 and −6.4 kcal.mol^−1^) compared to the affinity exhibited by Doxorubicin (−7.4 kcal.mol^−1^), a commercial anticancer drug, and chloroquine (−5.4 kcal.mol^−1^), a commercial antimalaria drug. Approximately 60 different *P. falciparum* erythrocyte membrane protein 1 (PfEMP1) variants are encoded by each haploid genome of *P. falciparum*. Such *P. falciparum* cerebral malaria (CM) parasites express a subgroup of group A PfEMP1s that facilitate the dual binding to host intercellular adhesion molecule-1 (ICAM-1) and endothelial protein C receptor EPCR [67]. The high affinity of mannotriose to this protein was predicted to inhibit *P. falciparum* growth.

Other studies related to molecular docking with *Plasmodium* protein have shown varying results and have been determined by the type of compound. Atorvastatin, traconazole, and posaconazole produce a binding energy ranging from −1.9 to −6.5 kcal/mol on *P. falciparum* protein lactate dehydrogenase (*Pf*LDH) [68], lower than that observed in this study. Thus, the mannotriose detected from marine bacterium *S. sennicomposti* GMY01 has potential as an anti-plasmodium, especially as a phosphoethanolamine methyltransferase (PMT) and erythrocyte membrane protein 1 (EMP1) inhibitor.

These two compounds have a higher affinity to autophagy proteins mTORC1 and mTORC2 than apoptosis proteins BCL2 and BCL XL (Table 5). Mannotriose, and N-acetylneuraminyl-(2,6)-galactose have a higher affinity to autophagy proteins. These results were similar to the anticancer activity of *S. sennicomposti* GMY01 on the lung cancer cell line in the previous study [69]. The analysis of protein expression using Western blot after treatment with GMY01 extract showed that the LC3-I and LC3-II autophagy-related protein expressions were higher than PARP apoptosis-related proteins [69].

Inhibition of the mTORC1 protein is associated with enhanced autophagy. However, another study showed the silencing of mTORC2-induced apoptosis [70]. This indicated that the autophagy inhibitor compounds could be used as an alternative drug candidate for anticancer. These results were similar to another study, which showed that mTORC1 and mTORC2 inhibitor compounds were used as models in the design and development of lung cancer treatments [71]. When mTORC1 is inhibited, the protein complex associated with autophagy activation recruits LC3 protein, which will be converted into active cytosolic isoform LC3-I and transformed into LC3-II. LC3 II protein is located in the autophagosome membrane and allows for the binding of degraded substrates. Autophagosomes will fuse with lysosomes and form autolysosomes, which play a role in the autophagy process [52]. In this study, mannotriose affinity (from −7.4 to −8.4 kcal.mol^−1^) and N-acetylneuraminyl-(2,6)-galactose affinity (from −6.3 to −8 kcal.mol^−1^) both on apoptosis and autophagy proteins was lower than doxorubicin (from −7.2 to −10.1 kcal.mol^−1^) as reference compounds for anticancer.

## 5. Conclusions

The whole-genome sequence analysis revealed that *Streptomyces* sp. GMY01 (7.9 Mbp) was most similar to *Streptomyces sennicomposti* RCPT1-4^T^. The bioactive compounds isolated from marine-derived actinobacteria GMY01 biomass exhibited moderate activity against *Plasmodium falciparum* FCR3 and human cervical carcinoma HeLa cell line, with low toxicity for Vero cells as normal cells. LC-MS/MS analysis revealed that the methanolic fraction of *S. sennicomposti* GMY01 biomass contained carbohydrate-based compounds, N-acetylneuraminyl-(2-6)-galactose (C_17_H_29_NO_14_) as a major compound, and mannotriose (C_18_H_32_O_16_) as a minor compound. This indicates that *S. sennicomposti* GMY01 is bacterium with potential carbohydrate-based bioactive anti-plasmodial and anticancer activity.

## 6. Patents

Indonesian Patent No. P00202109840 and No. IDP000082901.

## Figures and Tables

**Figure 1 microorganisms-11-01930-f001:**
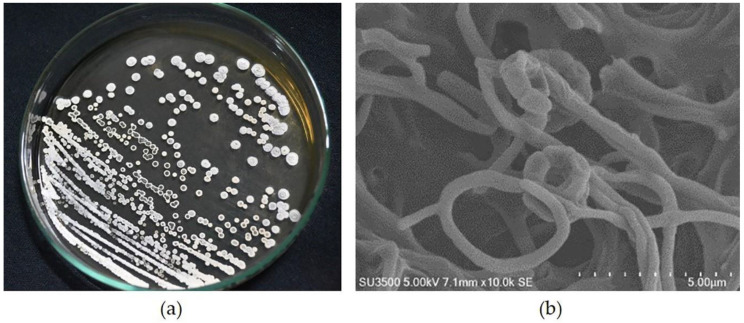
(**a**) Colony morphology of GMY01 in International *Streptomyces* Project-2; (**b**) scanning electron micrograph of isolated GMY01 on ISP-2 after 14 days at 28 °C, showing open spiral chains of smooth surface spores.

**Figure 2 microorganisms-11-01930-f002:**
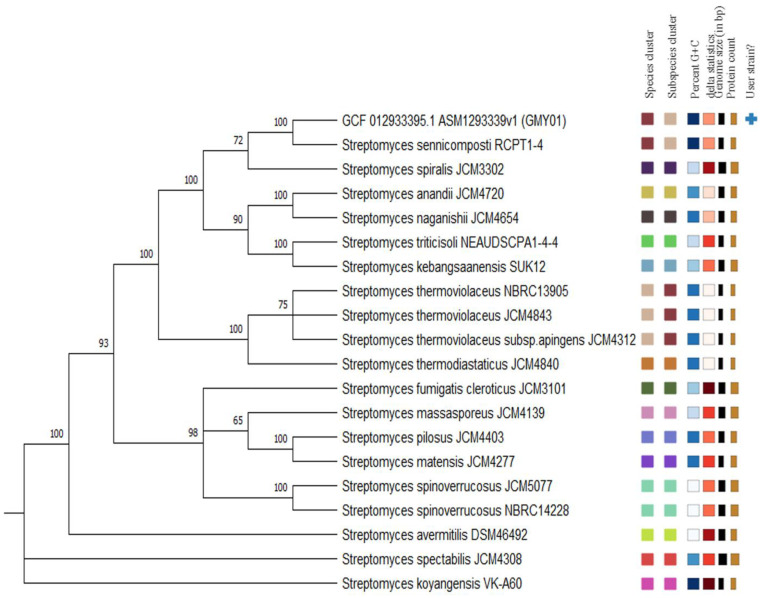
Phylogenic tree inferred using FastME 2.1.6.1 from Genome BLAST Distance Phylogeny (GBDP) calculated from genome sequences of strain GMY01 and other related genomes using online database Type Strain Genome Server (TYGS). The branch lengths are scaled in terms of GBDP distance formula d5. The numbers above branches are GBDP pseudo-bootstrap support values >60% from 100 replications, with an average branch support of 87.8%. The square labels colored by the same color indicate the genomes with the same species and subspecies gene clusters, percent G+C content (white), delta statistics, genome size and protein count.

**Figure 3 microorganisms-11-01930-f003:**
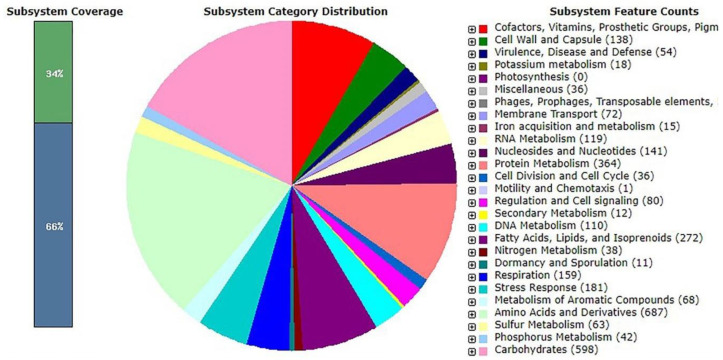
Overview of *Streptomyces* sp. GMY01 subsystem gene functions as generated by analysis using the RAST server at (http://rast.nmpdr.org (accessed on 17 January 2020)).

**Figure 4 microorganisms-11-01930-f004:**
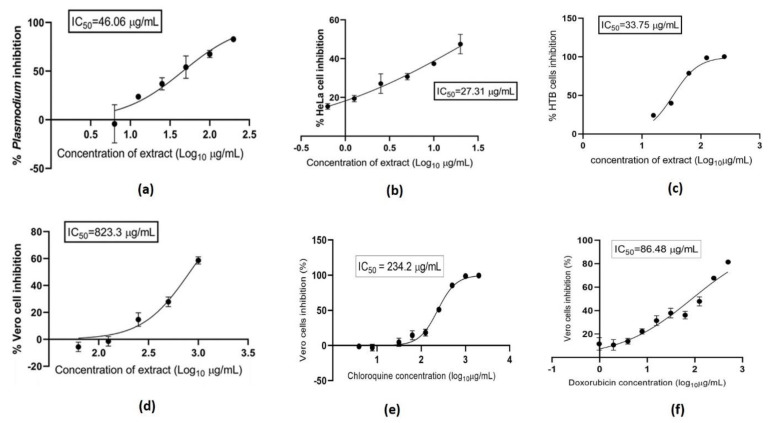
Biological activities of methanolic active fraction from marine *Streptomyces sennicomposti* GMY01 on *Plasmodium falciparum* FCR3 (**a**), HeLa cell line (**b**), HTB cell line (**c**), and Vero cell line (**d**). Toxicity of commercial anti-plasmodial—antimalarial (chloroquine) on Vero cell line (**e**) and Commercial anticancer (doxorubicin) on Vero cell line (**f**).

**Figure 5 microorganisms-11-01930-f005:**
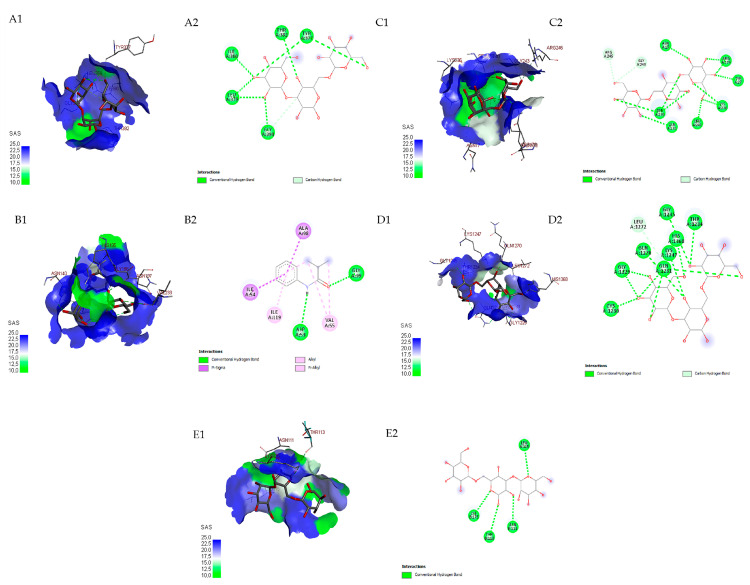
Three-dimensional docking visualization of mannotriose on *Plasmodium falciparum* proteins: GR: glutathione reductase (**A1**), LDH: lactate dehydrogenase (**B1**), PMT: phosphoethanolamine methyltransferase (**C1**), EMP1: erythrocyte membrane protein 1 (**D1**) and GST: glutathione-S-transferase (**E1**). Two-dimensional docking visualization of mannotriose on *P. falciparum* proteins: GR (**A2**), LDH (**B2**), PMT (**C2**), EMP1 (**D2**) and GST (**E2**).

**Figure 6 microorganisms-11-01930-f006:**
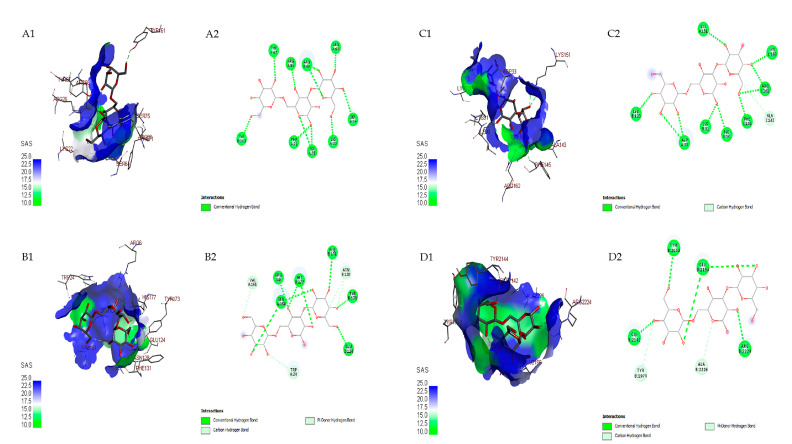
Three-dimensional docking visualization mannotriose on cancer cell proteins: BLC-2: B-cell lymphoma (**A1**); BCL-XL: B-cell lymphoma-extra-large (**B1**); mTORC1: mammalian target of rapamycin-1 (**C1**); mTORC2: mammalian target of rapamycin-2 (**D1**). Two-dimensional docking visualization mannotriose on cancer cell proteins: BLC-2 (**A2**); BCL-XL (**B2**); mTORC1 (**C2**); mTORC2 (**D2**).

**Table 1 microorganisms-11-01930-t001:** Pairwise comparisons of *Streptomyces* sp. GMY01 genome with the closest references genomes.

References	Size	Contigs	GC (%)	Ortho ANIu ^1^	Average Aligned Length (bp)	ANIb ^2^	Aligned (%)	z-Score ^3^
*S. sennicomposti* RCPT1-4^T^	7,351,330	150	73.09	98.09	4,709,881	97.33	70.86	0.99978
*S. spiralis* JCM 3302	9,813,663	176	71.26	86.15	3,378,116	85.75	50.63	0.98819
*S. kebangsaanensis* SUK12	8,244,886	170	71.57	85.61	2,923,156	84.66	44.98	0.98559
*S. naganishii* JCM 4654	7,809,671	76	72.61	85.73	3,120,830	85.02	48.11	0.98964

^1^: Analysis of average nucleotide identity (ANI) was performed using the ANI calculator (https://www.ezbiocloud.net/tools/ani, accessed on 18 June 2023). ^2^: The ANI based on BLAST+, ^3^: correlation indexes of tetra nucleotide signatures (TETRA) using the JSpecies software tool version 4.0.2 (https://jspecies.ribohost.com/jspeciesws/#analyse, accessed on 18 June 2023).

**Table 2 microorganisms-11-01930-t002:** Genome mining analysis of *Streptomyces sennicomposti* GMY01 using antiSMASH version 7.0.0.

Region	Type	Most Similar Known Cluster	Similarity (%)
1.1	T3PKS	Flaviolin, polyketide	100
1.2	NRPS, NAPAA	Stenothricin, NRP:cyclic depsipeptide	13
3.1	NI-siderophore	Grincamycin, polyketide type II	5
4.1	Terpene	Geosmin, terpene	100
4.2	NRP-metallophore, NRPS	Griseobactin, NRP	61
4.3	T1PKS, NRPS like, NRPS, butyrolactone, LAP	Microansamycin, Polyketide	66
4.4	NI-siderophore		
6.1	T2PKSs	Spore pigment, polyketide	83
16.1	Terpene	Isorenieratene, terpene	75
17.1	Lanthipeptide-class-iv	Venezuelin, RiPP:Lanthipeptide	50
17.2	RiPP-like		
17.3	NRPS, other, NRPS-like	S56-p1, NRP	17
18.1	Terpene	Albaflavenone	100
19.1	Butyrolactone	Gaburedin A-F, other	70
26.1	hgIE-KS, T1PKS	Vazabitide A, NRP	17
26.2	RiPP-like, lanthipeptide-class-iii	Informatipeptin, RiPP-lanthipeptide	100
28.1	Ectoin	Ectoin, other	100
34.1	NRP- metallophore, NRPS	Mirubactin, NRPS	50
36.1	NRP-metallophore, T1PKS, NRPS	Scabichelin, NRP	90
38.1	NRPS-like, NRPS	Antipain, NRP	100
39.1	NRPS	Saframycin A/B, NRP	12
47.1	NRPS, lanthipeptide-class-ii	Lysocin, NRP	9
48.1	Terpene	Hopene, terpene	61
53.1	Lanthipeptide-class-iv	Class IV lanthipeptide/SflA, RiPP	100
56.1	Butyrolactone		
57.1	T1PKSA	Abyssomicin M-X, polyketide	56
63.1	NRPS	Saframycin A/B, NRP	12
65.1	NRPS-like	Saframycin Mx1, NRP	66

T1PKS = Type I Polyketide Synthase; T2PKS = Type II Polyketide Synthase; T3PKS = Type III Polyketide Synthase; NRPS = Nonribosomal Polyketide Synthase; NAPAA = non-alpha poly-amino acids like e-Polylysin; NI-siderophore = NRPS-independent, IucA/IucC-like siderophores; LAP = Linear azol(in)e-containing peptides; RiPP-like = other unspecified ribosomally synthesized and post-translationally modified peptide product (RiPP) hgIE-KS = heterocyst glycolipid synthase-like PKS.

**Table 3 microorganisms-11-01930-t003:** Chemical composition of the polar active fraction of marine *Streptomyces sennicomposti* GMY01, detected using LC-MS/MS analysis.

Observed RT (min)	Compound	Observed *m*/*z*	Neutral Mass (Da)	Relative Abundance (% Area)
1.06	C_18_H_32_O_16_	527.1579	504.16903	1.96
1.08	C_17_H_29_NO_14_	472.1660	471.15880	97.50
1.64	C_19_H_31_NO_13_	482.1864	481.17954	0.54

**Table 4 microorganisms-11-01930-t004:** The binding affinity of detected compounds in methanol of *Streptomyces sennicomposti* GMY01 on potential binding domains of *Plasmodium falciparum*.

Compound	Target Proteins	Affinity	RMSD		Binds to Amino Acids
			Lower Bond	Upper Bond	
Mannotriose	GR (1ONF)	−7.3	1.834	2.645	Ile (380), Leu (379), Gly (394), Thr (382), Tyr (377)
	LDH (1CET)	−7.3	1.486	5.080	Val (233), Arg (109), Asn (140), His (195), Asn (197), Gly (196)
	PMT (4FGZ)	−7.7	1.834	6.370	Arg (246), Gly (243), Asn (17), Arg (179), Tyr (27), Lys (236), Leu (240), Ser (239), Gln (178)
	EMP1 (3CPZ)	−7.6	1.968	6.492	Leu (1272), Gly (1245), Thr (1234), His (1368), Lys (1247), Gln (1231), Gln (1270), Gly (1229), Cys (1230)
	GST (4ZXG)	−6.4	1.763	3.665	Leu (115), Asn (111), Thr (113), Lys (175)
N-acetylneuraminyl-(2-6)-galactose	GR (1ONF)	−6.7	1.961	3.964	Gly (394), Ile (380), Leu (379), Lys (213), Gln (351)
	LDH (1CET)	−6.6	1.577	2.613	Arg (109), Leu (237), Arg (171), Thr (232), Asp (168), Val (233), His (195), Val (138), Gly (196)
	PMT (4FGZ)	−6.5	1.933	3.334	Arg (179), Asp (242), Gln (178), Lys (236), Ser (239)
	EMP1 (3CPZ)	−6.4	1.770	4.672	Cys (1273), Cys (1230), Gln (1231), Arg (1228), Ile (1227), His (1368), Thr (1234), Gln (1270), Tyr (1366), Lys (1247), Asn (1271)
	GST (4ZXG)	−6.1	1.067	2.008	Lys (175), Phe (110)
Chloroquine *	GR (1ONF)	−5.1	1.762	2.566	Ala (480), Ala (472), Met (470), Ala (488), Ile (483), Thr (480)
	LDH (1CET)	−5.0	1.936	2.792	Val (233), Leu (201), Glu (310)
	PMT (4FGZ)	−4.9	1.891	2.856	Glu (28), Lys (177), Lys (25)
	EMP1 (3CPZ)	−5.4	1.917	2.932	Phe (1553), Tyr (1565), Ile (1512), Lys (1516)
	GST (4ZXG)	−5.5	1.884	2.547	Lys (46), Leu (180), Pro (177), Phe (183), Leu (180)
Doxorubicin **	GR (1ONF)	−7.8	1.684	2.170	Ala (399), Ser (396), Lys (213), Thr (392), Gly (394), Thr (382), Ile (380), Phe (214), Tyr (185)
	LDH (1CET)	−7.5	1.928	5.934	Ile (31), Ser (245), Pro (246)
	PMT (4FGZ)	−9.5	1.622	2.018	Asp (250), Arg (246), Asp (242), Tyr (175), Gly (243), Arg (179), Arg (97)
	EMP1 (3CPZ)	−7.4	1.810	2.307	His (1368), Val (1274), Gln (1231), Gly (1229), Cys (1230), Lys (1247)
	GST (4ZXG)	−7.4	1.656	2.103	Lys (175), Lys (110), Phe (110)

* The reference to an anti-plasmodial compound; ** the reference to an anticancer compound. Affinity (Gibbs energy (kcal.mol^−1^)). RMSD = Root Mean Square Deviation. GR: glutathione reductase, LDH: lactate dehydrogenase, PMT: phosphoethanolamine methyltransferase, EMP1: erythrocyte membrane protein 1, GST: glutathione-S-transferase.

**Table 5 microorganisms-11-01930-t005:** The binding affinity of detected compounds in methanol of *Streptomyces sennicomposti* GMY01 on potential binding domains of cancer cells.

Compound	Target Proteins	Affinity	RMSD		Binds to Amino Acids
			Lower Bond	Upper Bond	
Mannotriose	BCL-2 (2w3l)	−8.4	1.878	5.304	Tyr (67), Tyr (161), Arg (26), Arg (66), Arg (68), Ser (64), Lys (22), Ser (75), Phe (71)
	BCL XL (2yxj)	−7.4	1.891	5.662	Val (161), Arg (6), His (177), Ser (164), Phe (131), Asn (128), Tyr (173), Glu (124), Trp (24)
	mTORC1 (6BT0)	−8.0	0.965	2.103	Lys (120), Asp (33), Lys (91), Ile (90), Phe (145), Ala (143), Arg (162), Lys (169), Lys (151)
	mTORC2 (6zwo)	−8.3	1.691	2.599	Tyr (2144), Gly (2142), Tyr (1974), Ala (2226), Arg (2224), Glu (2196)
N-acetylneuraminyl-(2-6)-galactose	BCL-2 (2w3l)	−7.4	1.297	1.871	Asn (122), Arg (26), Lys (22), Ser (64), Arg (66), Arg (68), Arg (65), Ser (75), Phe (71)
	BCL XL (2yxj)	−6.3	1.962	6.445	Arg (6), Asn (128), Tyr (173), His (177), Ser (164)
	mTORC1 (6BT0)	−8.0	1.845	2.552	Ser (16), Tyr (35), Ser (34), Ser (20), Lys (19), Ser (21)
	mTORC2 (6zwo)	−7.2	1.929	2.840	Asp (594), Arg (363), Tyr (490), Leu (364), Trp (362), Ser (491), Gln (657), Leu (492), His (658), Lys (599)
Chloroquine *	BCL-2 (2w3l)	−5.8	1.904	5.152	Phy (71), Tyr (67), Met (74), Leu (96), Phe (63), Arg (105), Ala (108)
	BCL XL (2yxj)	−6.4	1.449	2.091	Ala (71), Phe (49), Phe (259), Leu (12), Asp (211), Asp (123)
	mTORC1 (6BT0)	−6.1	1.179	2.005	Ala (150), Lys (120), Phe (31), Asp (33), Pro (37)
	mTORC2 (6zwo)	−6.4	1.342	2.118	Ala (47), Cys (133), Ser (90), Thr (2279), Ala (89), Trp (274)
Doxorubicin **	BCL-2 (2w3l)	−10.1	1.429	2.034	Arg (68), Asp (61), Ser (64), Phe (71), Ser (75), Arg (66), Phe (63), Val (107)
	BCL XL (2yxj)	−7.2	1.474	2.413	Asn (136), Arg (139), Ala (142), Phe (97), Leu (130), Ala (104), Leu (108), Glu (129)
	mTORC1 (6BT0)	−9.2	1.577	2.214	Gly (29), Asp (33), Phe (31), Glu (139), Lys (135), Lys (91), Tyr (131)
	mTORC2 (6zwo)	−8.2	1.857	2.618	Glu (2190), Asp (2357), Ser (2165), Lys (2187), Ile (2537), Ile (2356), Try (2239), Thr (2245)

* The reference to anti-plasmodial compound; ** the reference to the anticancer compound. Affinity (Gibbs energy (kcal.mol^−1^)). RMSD = Root Mean Square Deviation. Apoptosis proteins = BLC2: B-cell lymphoma and BCL-XL: B-cell lymphoma-extra-large; autophagy proteins = mTORC1: mammalian target of rapamycin-1 and mTORC2: mammalian target of rapamycin-2.

## Data Availability

This whole-genome shotgun project was deposited at DDBJ/ENA/GenBank under the accession JABBNA000000000.

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
