# Peer review of "Marine-Derived Streptomyces sennicomposti GMY01 with Anti-Plasmodial and Anticancer Activities: Genome Analysis, In Vitro Bioassay, Metabolite Profiling, and Molecular Docking"

_microorganisms, 2023, doi:10.3390/microorganisms11081930_

Round 1

Reviewer 1 Report

Unfortunately, I can only evaluate part of this research.

1. From my point of view, the genomic analysis was performed very superficially. Annotation using RAST is not sufficient. Authors also must annotate this genome with other programs and servers. Describe genomic data at COG level. Authors must analyze the genome for the presence of putative gene clusters for the synthesis of secondary metabolites using at least a server antiSMASH. 

2. Phylogenetic analysis results are very outdated (May 2020, line 112). The data must be updated. Three years is a long time for the data to remain the same. ANI, AAI, dDDH values must be calculated again.

3. Lines 391-393. This information are very outdated (lpsn list, February 2020) and must be updated to June 2023.

4. The docking results (lines 332-339) should be transferred to an additional file in the form of a table, not as a text. There are Figure 5 and Figure 6, therefore authors should describe only the interaction of key residues and compare these interactions between the targets. The values ​of binding strength and/or bond length should be added in the table.

Reviewer 2 Report

In the manuscript “Marine Derived-Streptomyces sp. GMY01 with Anti-plasmodial and Anticancer Activities: Genome Analysis, Bioassay, Metabolite Profiling and Molecular Docking”, the authors present the discovery of new compounds from a novel actinomycete called Streptomyces sp. GMY01. Specifically, the authors compare the genome of Streptomyces sp. GMY01 with other Streptomyces species and suggest that is a new strain. The crude extract of this strain contains two compounds active against Plasmodium falciparum and diverse cancer cells. The authors identified compounds with anti-malarial and anti-cancer activity with low toxicity on normal cells. Although the topic is interesting, the manuscript needs some improvements before being accepted for publication.

English is very poor, and some words are too frequently used, i.e. especially, and the language should be carefully checked by an English speaker or English mother tongue; the authors should pay more attention to the correctness of some sentences along with the whole manuscript. The words Actinobacteria and Actinomycetales are improperly used.

Introduction

Major comments

Line 46, a reference should be added for the sentence: “It has a very large genome size, between 6.2 and 12.7 Mb”.

Line 57-60, This part is not very clear.

Lines 60-62, what do the authors mean by synergetic? It is not clear which other molecules they refer to.

Line 65-66, it is not clear, again, which molecules they refer to.

Lines 77-78, was the strain isolated in this study? If yes, the aim should be rewritten. If not, the authors should provide more details.

Material and Methods:

This section lacks information on the collection of the sediment, bacterial isolation, and DNA extraction.

Line 106-107, The link is not available.

Line 107-108, did the authors consider using an ANI cut-off of 97% since many of the actinomycetes’ genes are conserved or at least show high similarity?

Line 114, the authors should write the words of the abbreviation GBDP, for the first time. The authors should check all the abbreviations in the manuscript.

Line 209, were instead of was

 Results

Line 275, why did the authors incubate the strain for 11 days? Did the authors perform a growth curve followed by an anti-malarian and anti-tumoral profile of the extracts?

Line 236, the authors should rephrase the sentence “The colony will form 235 a pellet formation”.

Line 234-238, it is not clear if the strain was isolated in this study and the sequencing of the 16S rDNA was performed to identify the strain.

Line 268-270, the authors should rephrase the two sentences.

Line 282, the words “with IC50 was 823.3 282 µg/mL” should be replaced by “with an IC50 of 823.3 282 µg/mL”

Line 295-296, the sentence “Mannotriose that was detected both in genome mining and MS analysis” lacks the main verb.

The legend of all the Figures, mainly 3 and 4, should provide more details.

Line 266, which strains do the authors refer to in the sentence “Taxonomic identification of 266 query strains”?

Lines 268-270: The authors should rephrase the sentences.

Discussion

In some parts, the authors repeat their results, making the discussion more protracted than necessary.

Lines 522-528: it is not clear if the authors report their results or from other studies.

Line 393-401, the authors are suggested to re-write the sentences.

Line 413, the authors should be consistent with the name of the compounds along the text.

Line 511, what does CM refer to?

Line 548 the authors should remove µg/mL)?

English is very poor, and some words are too frequently used, i.e. especially, and the language should be carefully checked by an English speaker or English mother tongue; the authors should pay more attention to the correctness of some sentences along with the whole manuscript. The words Actinobacteria and Actinomycetales are improperly used.

Round 2

Reviewer 1 Report

I still have serious remarks about the results and the text of the manuscript.

Streptomyces sennicomposti RCPT1-4T is validly published under the ICNP and has a correct name (https://lpsn.dsmz.de/species/streptomyces-sennicomposti). Based on genome indices, such as ANI and ANIb, authors must add this species name in the manuscript title: “Marine Derived-Streptomyces sennicomposti GMY01 with Anti-plasmodial and Anticancer Activities: Genome Analysis, In Vitro Bioassay, Metabolite Profiling, and Molecular Docking”. Also, authors should replace ‘sp’ with ‘sennicomposti’ throughout the text after Line 262.

Line 40. Start the sentence with ‘Marine actinobacteria’.

Line 47. Replace ‘especially’ with ‘mainly’.

Line 55. Must be ‘drug-resistant Plasmodium spp. drive'.

Line 58. Add ‘drug’ or ‘activity” after ‘antimalarial’. It is an adjective.

Line 61. Add ‘one’ after ‘antimalarial’, because it is an adjective. Check, please, throughout the text.

Line 62. Why does a word ‘Inhibitor’ start with a capital letter?

Line 97. Replace ‘taxogenomic’ with ‘taxonogenomic’.

Line 99. Replace, please, ‘and’ with ‘such as’.

Line 116. Replace ‘strain types’ with ‘type strains’.

Line 248. Why did the authors write ‘Illumina methods’, if they wrote ‘454 GS FLX’ on the Methods (lines 98-100)? This is not Illumina technology. Please, correct text on this line.

Line 254. On the Figure 3, there are 16 species groups, not 20 ones.

Line 257. Authors have to use a value of 82,3 from formula d4 because “it is immune against problems caused by incompletely sequenced genomes, as it does not consider the genome lengths” (Auch AF, et al. 2010, doi: 10.4056/sigs.531120).

Line 260. Must be ‘reference strains’.

Line 267. Must be ‘with the closest reference genomes’. Replace ‘OrthuANIu” with ‘OrthoANIu’. Delete a word ‘strain’.

Line 313. Delete a word ‘protein’.

Lines 312-313. The manuscript does not provide data to support this claim “Mannotriose was detected both in genome mining…”.

Line 334. Delete a word ‘protein’.

Lines 371-372. Delete ‘strain’ and ‘bacterium’.

 Lines 374-379. It is completely unclear why the authors discussed algorithms that were not used in their study. If they have used OrthoANIu and ANIb, then they should discuss and cite the following articles:

(OrthoANIu) Yoon SH, Ha SM, Lim J, Kwon S, Chun J. A large-scale evaluation of algorithms to calculate average nucleotide identity. Antonie Van Leeuwenhoek. 2017 Oct;110(10):1281-1286. doi: 10.1007/s10482-017-0844-4. Epub 2017 Feb 15. PMID: 28204908.

(ANIb) Richter M, Rosselló-Móra R. Shifting the genomic gold standard for the prokaryotic species definition. Proc Natl Acad Sci U S A. 2009 Nov 10;106(45):19126-31. doi: 10.1073/pnas.0906412106. Epub 2009 Oct 23. PMID: 19855009; PMCID: PMC2776425.

Lines 393-399. Move the text to Section “Main constituent profile”. Authors should provide more genome information on sugar biosynthesis, in particular mannotriose and N-acetylneuraminyl-(2-6)-galactose. Supplementary Fig. S3 does not support the authors' claim of galactose metabolism. The scheme should show the presence of genes in green color, at least. Rename this Section as “Main constituent profile and predicted galactose metabolic pathway”.

Line 433. Add references to the statement “However, several studies reported that mannotriose is a compound that composes one of the important prebiotics, namely mannooligosaccharides (MOS), and is predicted to have anticancer activity.”

Line 685. A name of a journal is absent.

A very serious editing of the English language and style is required. Lots of inaccuracies and typos. Nouns are missing.

For example:

Lines 33-35. Streptomyces sp. GMY01 is a potential bacterium producing carbohydrate-based bioactive as anti-plasmodial and anticancer with low toxicity on normal cells.

Should be:

Streptomyces sennicomposti GMY01 is a potential bacterium producing carbohydrate-based bioactive compounds with anti-plasmodial and anti-cancer activities and with low toxicity to normal cells.

Author Response

Reviewer 1

  1. I still have serious remarks about the results and the text of the manuscript.

Streptomyces sennicomposti RCPT1-4is validly published under the ICNP and has a correct name (https://lpsn.dsmz.de/species/streptomyces-sennicomposti). Based on genome indices, such as ANI and ANIb, authors must add this species name in the manuscript title: “Marine Derived-Streptomyces sennicomposti GMY01 with Anti-plasmodial and Anticancer Activities: Genome Analysis, In Vitro Bioassay, Metabolite Profiling, and Molecular Docking”. Also, authors should replace ‘sp’ with ‘sennicomposti’ throughout the text after Line 262.

Author Comment: Thank you for valuable comment and suggestion. We have revised manuscript according to the suggestions

  1. Line 40. Start the sentence with ‘Marine actinobacteria’

Author Comment: We have revised it according to the suggestions (line 40)

  1. Line 47. Replace ‘especially’ with ‘mainly’.

Author Comment: We have revised it according to the suggestions (line 47)

  1. Line 55. Must be ‘drug-resistant Plasmodium drive'.

Author Comment: We have revised it according to the suggestions (line 56)

  1. Line 58. Add ‘drug’ or ‘activity” after ‘antimalarial’. It is an adjective.

Author Comment: We have revised it according to the suggestions (line 58)

  1. Line 61. Add ‘one’ after ‘antimalarial’, because it is an adjective. Check, please, throughout the text.

Author Comment: We have revised it according to the suggestions (line 61)

  1. Line 62. Why does a word ‘Inhibitor’ start with a capital letter?

Author Comment: That is typo. We have replaced with “inhibitor” (line 62)

  1. Line 97. Replace ‘taxogenomic’ with ‘taxonogenomic’.

Author Comment: We have revised it according to the suggestions (line 97)

  1. Line 99. Replace, please, ‘and’ with ‘such as’.

Author Comment: We have revised it according to the suggestions (line 99)

  1. Line 116. Replace ‘strain types’ with ‘type strains’.

Author Comment: We have revised it according to the suggestions (line 116)

  1. Line 248. Why did the authors write ‘Illumina methods’, if they wrote ‘454 GS FLX’ on the Methods (lines 98-100)? This is not Illumina technology. Please, correct text on this line.

Author Comment: We have revised with “using a combined shotgun sequencing method with the 454 GS FLX Titanium system (Roche) and paired-end (PE) sequencing using the HiSeq 1000 platform (Illumina)” (line 98 – 100).

  1. Line 254. On the Figure 3, there are 16 species groups, not 20 ones.

Author Comment: We have revised with “16 species groups” (line 253)

  1. Line 257. Authors have to use a value of 82,3 from formula d4 because “it is immune against problems caused by incompletely sequenced genomes, as it does not consider the genome lengths” (Auch AF, et al. 2010, doi: 10.4056/sigs.531120).

Author Comment: We have revised it according to the suggestions (line 257)

  1. Line 260. Must be ‘reference strains’.

Author Comment: We have revised it according to the suggestions (line 259)

  1. Line 267. Must be ‘with the closest reference genomes’. Replace ‘OrthuANIu” with ‘OrthoANIu’. Delete a word ‘strain’.

         Author Comment: Thank you for the suggestion. We have revised it according to the suggestions (line 267)

  1. Line 313. Delete a word ‘protein’.

Author Comment: there is no word “protein” in that line

  1. Lines 312-313. The manuscript does not provide data to support this claim “Mannotriose was detected both in genome mining…”.

Author Comment: We have deleted that claim. Only gene clusters related to galactose detected in genome mining (Line 322 – 329)

  1. Line 334. Delete a word ‘protein’.

Author Comment: We have revised it according to the suggestions (line 341)

  1. Lines 371-372. Delete ‘strain’ and ‘bacterium’.

Author Comment: We have revised it according to the suggestions (line 379 – 380)

  1. Lines 374-379. It is completely unclear why the authors discussed algorithms that were not used in their study. If they have used OrthoANIu and ANIb, then they should discuss and cite the following articles:

(OrthoANIu) Yoon SH, Ha SM, Lim J, Kwon S, Chun J. A large-scale evaluation of algorithms to calculate average nucleotide identity. Antonie Van Leeuwenhoek. 2017 Oct;110(10):1281-1286. doi: 10.1007/s10482-017-0844-4. Epub 2017 Feb 15. PMID: 28204908.

(ANIb) Richter M, Rosselló-Móra R. Shifting the genomic gold standard for the prokaryotic species definition. Proc Natl Acad Sci U S A. 2009 Nov 10;106(45):19126-31. doi: 10.1073/pnas.0906412106. Epub 2009 Oct 23. PMID: 19855009; PMCID: PMC2776425.

Author Comment: Thank you for the suggestion. We have revised the references for discussion (line 383 – 386)

  1. Lines 393-399. Move the text to Section “Main constituent profile”. Authors should provide more genome information on sugar biosynthesis, in particular mannotriose and N-acetylneuraminyl-(2-6)-galactose. Supplementary Fig. S3 does not support the authors' claim of galactose metabolism. The scheme should show the presence of genes in green color, at least. Rename this Section as “Main constituent profile and predicted galactose metabolic pathway”.

Author Comment: The scheme was revised (Supplementary file Fig.3) and add color for genes and products. Section 3.3 was renamed as “Main constituent profile and predicted galactose metabolic pathway (Line 306). Genome information about sugar biosynthesis was provided in result (Line 325 – 327)

  1. Line 433. Add references to the statement “However, several studies reported that mannotriose is a compound that composes one of the important prebiotics, namely mannooligosaccharides (MOS), and is predicted to have anticancer activity.”

Author Comment: We have added supporting references in the text (line 429 – 430)

  1. Line 685. A name of a journal is absent.

Author Comment: The reference is the name of the program used, so there is no journal name

  1. Comments on the Quality of English Language

A very serious editing of the English language and style is required. Lots of inaccuracies and typos. Nouns are missing.

For example:

Lines 33-35. Streptomyces sp. GMY01 is a potential bacterium producing carbohydrate-based bioactive as anti-plasmodial and anticancer with low toxicity on normal cells.

Should be:

Streptomyces sennicomposti GMY01 is a potential bacterium producing carbohydrate-based bioactive compounds with anti-plasmodial and anti-cancer activities and with low toxicity to normal cells.

Author Comment: We have revised according to suggestions (Line 34 – 35). We have made improvements in English writing based on suggestions from language editing professionals

Reviewer 2 Report

In my opinion, the authors answered my concerns

In my opinion, the English of the newly added sentences could be improved.

Author Response

Reviewer 2

  1. Comments and Suggestions for Authors

In my opinion, the authors answered my concerns

Author comment: Thank you for receiving improvements from us

  1. Comments on the Quality of English Language

In my opinion, the English of the newly added sentences could be improved.

Author comment: We have made improvements in English writing based on suggestions from language editing professionals